



# 30 m annual land cover and its dynamics in China from 1990 to 2019

Jie Yang[1], Xin Huang[1,2]

[1] School of Remote Sensing and Information Engineering, Wuhan University, Wuhan, 430079, P.R. China
[2] State Key Laboratory of Information Engineering in Surveying, Mapping and Remote Sensing, Wuhan University, Wuhan,
430079, P.R. China

*Correspondence to*: Xin Huang (xhuang@whu.edu.cn)

**Abstract.** Land cover (LC) determines the energy exchange, water and carbon cycle between Earth's spheres. Accurate LC information is a fundamental parameter for the environment and climate studies. Considering that the LC in China has been altered dramatically with the economic development in the past few decades, sequential and fine-scale LC monitoring is in

urgent need. However, currently, fine-resolution annual LC dataset produced by the observational images is generally unavailable for China due to the lack of sufficient training samples and computational capabilities. To deal with this issue, we produced the first Landsat-derived annual China Land Cover Dataset (CLCD) on the Google Earth Engine (GEE) platform, which contains 30 m annual LC and its dynamics of China from 1990 to 2019. We first collected the training samples by combining stable samples extracted from China's Land-Use/Cover Datasets (CLUD), and visually-interpreted

samples from satellite time-series data, Google Earth and Google Map. Using 335,709 Landsat images on the GEE, several temporal metrics were constructed and fed to the random forest classifier to obtain classification results. We then proposed a post-processing method incorporating spatial-temporal filtering and logical reasoning to further improve the spatial-temporal consistency of CLCD. Finally, the overall accuracy of CLCD reached 79.31% based on 5,463 visually-interpreted samples. A further assessment based on 5,131 third-party test samples showed that the overall accuracy of CLCD outperforms that of

MCD12Q1, ESACCI_LC, FROM_GLC, and GlobaLand30. Besides, we intercompared the CLCD with several Landsat-derived thematic products, which exhibited good consistencies with the Global Forest Change, the Global Surface Water, and three impervious surface products. Based on the CLCD, the trends and patterns of China's LC changes during 1985 and 2019 were revealed, such as expansion of impervious surface (+148.71%) and water (+18.39%), decrease of cropland (-4.85%) and grassland (-3.29%), increase of forest (+4.34%). In general, CLCD reflected the rapid urbanization and a series

of ecological projects (e.g., Gain for Green) in China and revealed the anthropogenic implications on LC under the condition of climate change, signifying its potential application in the global change research. The CLCD dataset introduced in this article is freely  available at http://doi.org/10.5281/zenodo.4417810 (Yang and Huang, 2021).

## Introduction

Land cover (LC) is an essential component of the Earth System and closely connects the biosphere, atmosphere, and

hydrosphere. It is usually divided into a range of hierarchical categories, each providing unique habitats and determining the





energy exchange, water balances and carbon cycling (Gómez et al., 2016; Houghton et al., 2012; Tang, 2020; Wulder et al., 2018). LC is important for land surface processes simulation and is a key variable for environment and ecology models (Schewe et al., 2019; Wulder et al., 2018). In addition, as human settlements sprawled rapidly over the past few decades (Goldewijk, 2001), more demands were stressed on the terrestrial ecosystem goods and services (Friedl et al., 2010).

Consequently, anthropogenic activities have significant implications on LC, water cycling, air quality, food supply, and biodiversity (Leng et al., 2015; Li et al., 2020a; Xiao et al., 2018). Accurate and timely LC information is therefore immensely important for climate and environment studies (Herold et al., 2006; Yang et al., 2019), food security (Yang et al., 2020b), sustainable development (Dewan and Yamaguchi, 2009), and resource management (Goetz et al., 2003).

Satellite remote-sensing promotes an efficient LC monitoring by gathering long-term and high-resolution Earth Observation

(EO) data through orbiting platforms. So far, there have been a lot of studies focus on LC mapping using satellite data. For example, Sulla-Menashe et al., (2019) used the STEP (The System for Terrestrial Ecosystem Parameterization) global training set and long-term Moderate Resolution Imaging Spectroradiometer (MODIS) data to provide annual LC maps that spans 2001–2018 (MCD12Q1, 500 m). Besides, the European Space Agency Climate Change Initiative (ESACCI) made an effort to a global LC monitoring (ESACCI_LC, 300 m) based on multi-source EO data and machine learning (Defourny et

al., 2017). More recently, Liu et al., (2020) produced an annual global LC product for 1982–2015 using 5-km GLASS (Global Land Surface Satellite) data. Although these LC products have wide temporal span, their spatial resolution is relatively coarse, which is not sufficient for fine-scale LC monitoring. Furthermore, the uncertainty inherent to LC maps from coarse-resolution data may hinder our understanding on the time-series LC dynamics (Sulla-Menashe et al., 2019).

Recently, the free availability and accessibility of high-resolution EO data (e.g., Landsat) (Woodcock et al., 2008), enables

fine-scale LC monitoring at large-scale. In a pioneering effort, Gong et al., (2013) generated the first global 30 m LC map, FORM-GLC (Finer Resolution Observation and Monitoring of Global Land Cover), based on Landsat images and over 90,000 visually-interpreted training samples. More recently, Gong et al., (2019) used training samples derived from the FORM-GLC and multi-temporal Sentinel-2 images to produce a 10 m global LC map for 2017. More recently, Zhang et al., (2020) combined the Global Spatial Temporal Spectra Library and Landsat time-series to map 30 global LC types for 2015.

However, due to the massive volume of data and the difficulties in obtaining multi-temporal training samples, high-resolution annual or continuous time-series LC products at large-scale have rarely been investigated in the current literature. Specifically, Liu et al., (2014) generated a 30 m LC product (China's Land-Use/cover Datasets, CLUD) via human-computer interactive interpretation over Landsat images, which documented LC of China from 1980s to 2015 at an interval of 5 years. However, annual LC information is not available for CLUD due to the tremendous workload and intensive resources

involved.

For the past few decades, rapid economic development and population growth has brought about notable implications on LC in China (Yao and Zhang, 2001). Meanwhile, China has implemented a series of ecological projects since the 1980s, such as Gain for Green, Red Lines of Cropland, which have played an important role in LC changes (Lü et al., 2013; Yin and Yin, 2010). In addition, the climate changes such as frequent precipitation extremes and temperature fluctuations also influenced





the LC change of China (Lutz et al., 2014; Zhai et al., 1999). Besides, in the recent decades, rapid urbanization had triggered significant loss of cropland and water. In this context, an annual high-resolution LC product would give us essential insights into both anthropogenic influences and natural changes, and help policymakers to implement informed and sustainable management.

In summary, high-resolution annual LC maps for China is still absent. To address this issue, in this research, we produce the
annual China Land Cover Dataset (CLCD), which to the best of our knowledge is the first Landsat-derived annual LC product of China from 1990 to 2019. To achieve this, we first automatically derived samples from CLUD and incorporated it with our visually-interpreted samples to obtain multi-temporal training samples. On the other hand, in recent years, the Google Earth Engine empowers a paradigm shift from traditional per-scene analysis to per-pixel analysis, which enables us to obtain large-scale pixel-wise image composites (Azzari and Lobell, 2017). Therefore, using all available Landsat images
(335,709) on the GEE, we calculated a set of spectral, phenological and topographical metrics via pixel-wise temporal composite. Subsequently, we generated CLCD by combining multi-temporal training samples, Landsat-derived temporal metrics and random forest (RF) classifier. Besides, to enable the LC change monitoring backdate to 1985, we generated a LC map for 1985 as a supplement to the CLCD. Lastly, a spatial-temporal post-processing method involving the spatial-temporal filter and logical reasoning was proposed to ensure the consistency of CLCD. The accuracy of CLCD was validated
by two open-source test sets and a visually-interpreted test set. In addition, we performed inter-comparison with thematic-class products (i.e., water, forest and impervious surface) to better reflect the quality of CLCD. Based on CLCD, we further analysed the trend of LC changes and conversions in China over the past four decades.

## 2 Data

### 2.1 Satellite data

Landsat satellites have been collecting 30 m EO data since the launch of Landsat 5 in 1984, which has been widely recognized as ideal data sources for high-resolution and large-scale LC monitoring. Thus, based on all available Landsat surface reflectance (SR) data on the GEE, we calculated input features including spectrum, phenology and topography. Clouds and cloud shadows in the SR data were identified and removed by the CFmask algorithm (Zhu and Woodcock, 2012). The systematic atmospheric and terrain correction have been conducted for Landsat SR data from all the sensors, i.e., the
Thematic Mapper (TM), the Enhanced Thematic Mapper Plus (ETM+), the Operational Land Imager (OLI), by the United States Geological Survey (USGS). However, given the inconsistency between different Landsat sensors, we used only Landsat 8 OLI data for the CLCD after 2013 and combined TM and ETM+ data before 2013 in view of the good spectral consistency between the two sensors (Micijevic et al., 2016). Due to the uneven spatial coverage of Landsat 5 data in China before 1990 (Pekel et al., 2016), we used all images captured before 1990 to generate the CLCD for the nominal year of
1985 and the CLCD after 1990 was produced annually.

In addition, slope and aspect were computed from the Shuttle Radar Topography Mission (SRTM) data to better reflect topographic changes and detect the LCs growing on steep slopes. Geographic coordinates (i.e., latitude and longitude) were also selected as input data, considering that the distribution of LCs is related to their geographic location.

## 2.2 China's Land-Use/cover Datasets

The China's Land-Use/cover Datasets (CLUD) documented detailed LC in China for 1980s, 1990, 1995, 2000, 2005, 2010, and 2015. CLUD was produced by human-computer interaction interpretation over Landsat images, consisting of 6 level-1 classes (cropland, forest, grassland, water, built-up area and barren) and 25 level-2 classes (Liu et al., 2003, 2014). Assessed via field survey, the overall accuracy of CLUD was reported higher than 94.3% for level-1 classes and more than 91.2% for level-2 classes (Liu et al., 2014). Although CLUD was provided at a 5-year interval, its unchanged area can be used as

potential training samples. In this study, we therefore proposed to automatically collect training samples via CLUD data for our 30 m annual LC mapping.

## 2.3 Third-party validation samples

In addition to the visually-interpreted test samples (see Sect. 3.5), we employed two third-party test sample sets to comprehensively validate the quality of CLCD. The first was Geo-Wiki (Fritz et al., 2017), which was a crowdsourced test

set covering 10 major LCs (Table S1). Based on the quality flag, we selected 3,000 "Quite Sure" Geo-Wiki samples located in China. The other was the Global Land Cover Validation Sample Set (GLCVSS) (Zhao et al., 2014), which followed an random sampling strategy to ensure even distribution of test samples at global scale. The classification system of the GLCVSS was the same with FROM-GLC (Table S1). We selected 2,131 GLCVSS samples covering China to assess the accuracy of CLCD.

## 115 2.4 Existing annual LC products

We intercompared the CLCD with the MCD12Q1 and ESACCI_LC to better reflect its quality. The MODIS land cover product (MCD12Q1) in Collection 6 was obtained using a supervised classification method (Sulla-Menashe et al., 2019), which provided global LC from 2001 to 2018 at 500 m resolution. Considering the comparability with the CLCD, the International Geosphere-Biosphere Program (IGBP) layer in MCD12Q1 was selected and remapped to the CLCD

classification system (Table S1). The ESACCI_LC was produced via the GlobCover unsupervised classification chain and multi-source EO data (Bontemps et al., 2013), which documented 300 m global LC during 1992–2018. Likewise, we remapped the class label of ESACCI_LC (Table S1) to facilitate the inter-comparison.



## 3 Method

This study was aimed at developing the CLCD dataset, and the processing chain included generation of training and test
samples, construction of annual input features, classification and spatial-temporal consistency check, as well as accuracy
assessment and products inter-comparison (Fig. 1). The procedure was implemented on the GEE platform, which enabled us
to perform pixel-wise analysis and freed us from data download and management (Gorelick et al., 2017). Public data on the
GEE, such as Landsat and MODIS, provided long-term Earth observations that helped us composite temporal metrics and
collect samples for training and validation. Finally, the accuracy of CLCD was evaluated by the visually-interpreted
independent samples and the third-party test samples. In particular, we intercompared CLCD with the current state-of-the-art
30 m thematic products including impervious surface area (ISA), surface water and forest to comprehensively assess the
quality of CLCD.

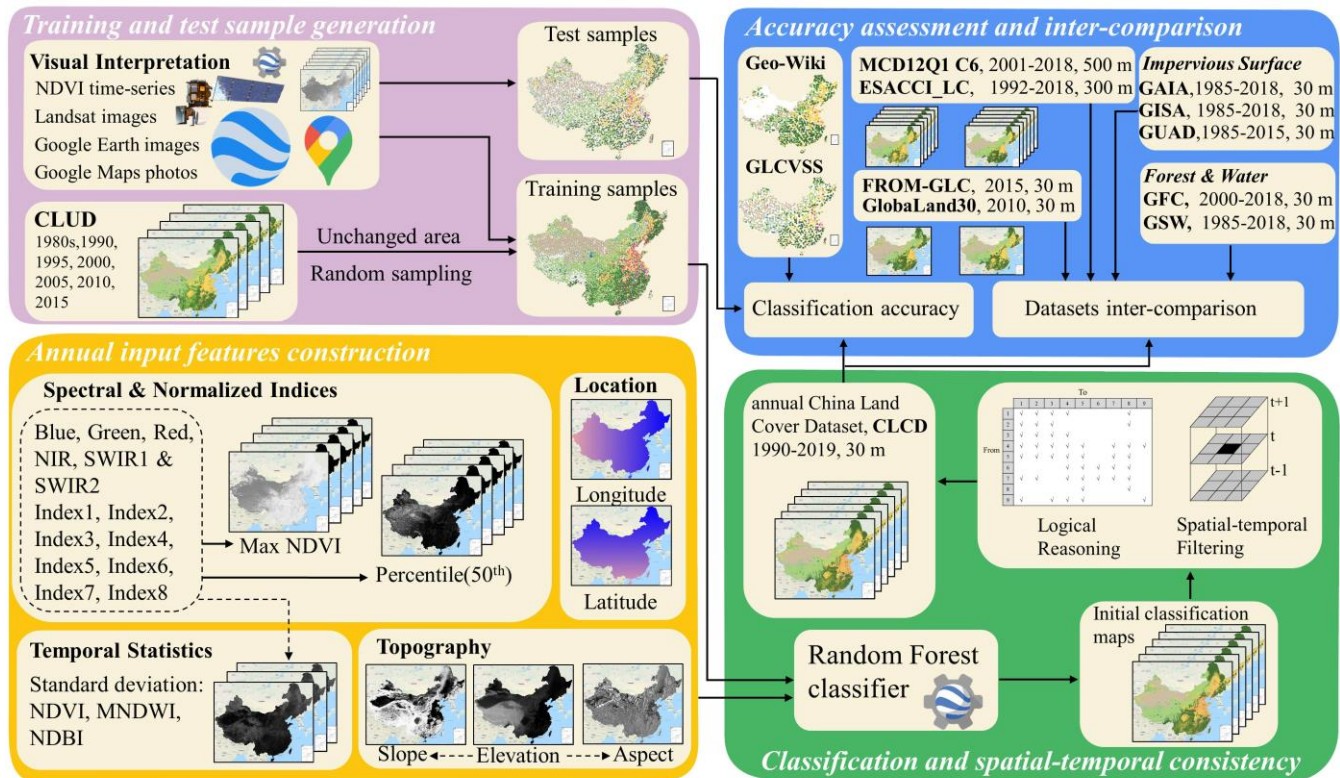

**Figure 1. The flow chart to generate the CLCD (annual China Land Cover Dataset).**

**3.1 Classification system**

Considering the LC distribution in China (Liu et al., 2018), we defined a classification system including 9 major LCs:
cropland, forest, shrub, grassland, water, snow and ice, barren, impervious, and wetland. This classification system is similar



to that of FROM-GLC (Gong et al., 2013) and can be conveniently remapped to the FAO (Food and Agriculture Organization) and IGBP systems.

## 3.2 Input features

The input features to the RF classifier were calculated in terms of spectrum, spectral index, phenology and geographic location (Table 1). Firstly, based on all available Landsat SR within a target year, we calculated the 50th percentile value for each spectral band. Given that spectral indexes can effectively enhance the difference among different LCs (Li et al., 2019), we computed eight spectral indexes to improve the discrimination ability of LC (Table 1). The spectral indexes were mainly constructed by two short-wave infrared (SWIR) bands (e.g., Band 5 and 7 for Landsat 5), since SWIR bands have better capability of atmospheric transmission. Besides, as suggested by Li et al., (2020b), to better distinguish between vegetation and non-vegetation, the spectral values of Landsat images corresponding to the maximum NDVI (Normalized Difference Vegetation Index) (Tucker, 1979) were included in the spectral features (e.g., Blue_NDVIMax in Table 1), and these spectral values were also used to calculate the aforementioned spectral indexes. Considering the spectral features of different LCs (e.g., vegetation, water) varied throughout the year, we also calculated the standard deviation of these spectral indexes, e.g., NDVI, MNDWI (Modified Normalized Difference Water Index) (Xu, 2006), NDBI (Normalized Difference Built-up Index) (Zha et al., 2003) to further highlight the phenological information. In summary, 36 features were obtained, including 12 spectral bands, 16 normalized spectral indices, 3 temporal statistics, 3 topographic features, and 2 geographical coordinates (Table 1). This approach of using all available images enabled us to: 1) reduce the dimension of input features while preserving temporal information, and 2) minimize the effects from clouds, shadows, or other disturbance.

**Table 1. The explanatory table of features used for CLCD mapping***

| Type | Features | Description | Resolution | Dimension | Source |
|---|---|---|---|---|---|
| Spectrum | Red, Green, Blue, NIR, SWIR1 & SWIR2; Blue_NDVIMax, Green_NDVIMax, Red_NDVIMax, NIR_NDVIMax, SWIR1_NDVIMax, SWIR2_NDVIMax; | 50th percentile value of surface reflectance derived from all available images within a study period, and surface reflectance of the Landsat image with the maximum NDVI. | 30 m | 6*2 | Landsat |
| Spectral Indices | Index1, Index2, Index3, Index4, Index5, Index6, Index7, Index8; Index1_NDVIMax, Index2_NDVIMax, Index3_NDVIMax, Index4_NDVIMax, Index5_NDVIMax, Index6_NDVIMax, Index7_NDVIMax, Index8_NDVIMax; | Normalized Indices derived from the corresponding spectral bands. The indices are calculated as: $Index1 = (SWIR2- NIR) / (SWIR2+ NIR)$; $Index2 = (SWIR2- Red) / (SWIR2+ Red)$; $Index3 = (SWIR2- Green) / (SWIR2+ Green)$; $Index4 = (SWIR2- SWIR1) / (SWIR2+ SWIR1)$; $Index5 = (SWIR1- NIR) / (SWIR1+ NIR)$; $Index6 = (SWIR1- Red) / (SWIR1+ Red)$; $Index7 = (SWIR1- Green) / (SWIR1+ Green)$; $Index8 = (NIR- Red) / (NIR+ Red)$. | 30 m | 8*2 | Landsat |
| Temporal Statistics | NDVI_StdDev, MNDWI_StdDev, NDBI_StdDev; | Standard deviation of NDVI, MNDWI and NDBI. | 30 m | 3 | Landsat |
| Topography | Elevation, Slope & Aspect; | Slope and aspect calculated from the SRTM elevation. | 30 m | 3 | SRTM |
| Location | Latitude & longitude; | Longitude and latitude at each pixel. | 30 m | 2 | N/A |

* Red, Green, Blue, NIR, SWIR1 and SWIR2 represent the Landsat data in visible band, near-infrared band, and shortwave band, respectively. NDVI, MNDWI and NDBI are abbreviations for the normalized difference vegetation index, the modified normalized difference water index, and the normalized difference built-up index, respectively.

**3.3 Training sample generation**

In the case of supervised large-scale LC mapping, accurate and adequate training samples are immensely essential. (Foody and Arora, 1997; Foody and Mathur, 2004). Usually, the strategies of training sample collection for a large-scale mapping task include: 1) visually-interpreted samples and 2) samples automatically derived from existing LC products (Zhang et al., 2020b). The visual interpretation method can obtain high quality samples but require intensive human labour. Whereas the automatic sample extraction via existing LC products is potential for generating a mass of randomly distributed samples, but the sample quality is related to the products used (Jokar Arsanjani et al., 2016; Wessels et al., 2016). Accordingly, the aforementioned two methods were both used to collect training samples in this study. Firstly, given that CLUD yielded an overall accuracy over 90% and has been used in a number of studies (Liu et al., 2014), it was considered as a source of training samples. Specifically, we selected the regions with stable LC throughout all periods of CLUD (i.e., 1980s, 1990, 1995, 2000, 2005, 2010, 2015) to further ensure the reliability of samples. In such a way, we obtained a candidate sample





pool of China. Then, the study area was divided into 1,665 hexagons with half-degree sides (Fig. 2a), and 20 points were randomly generated within each grid to ensure their spatial distribution and diversity. Finally, a total of 27,000 trainings samples were randomly selected.

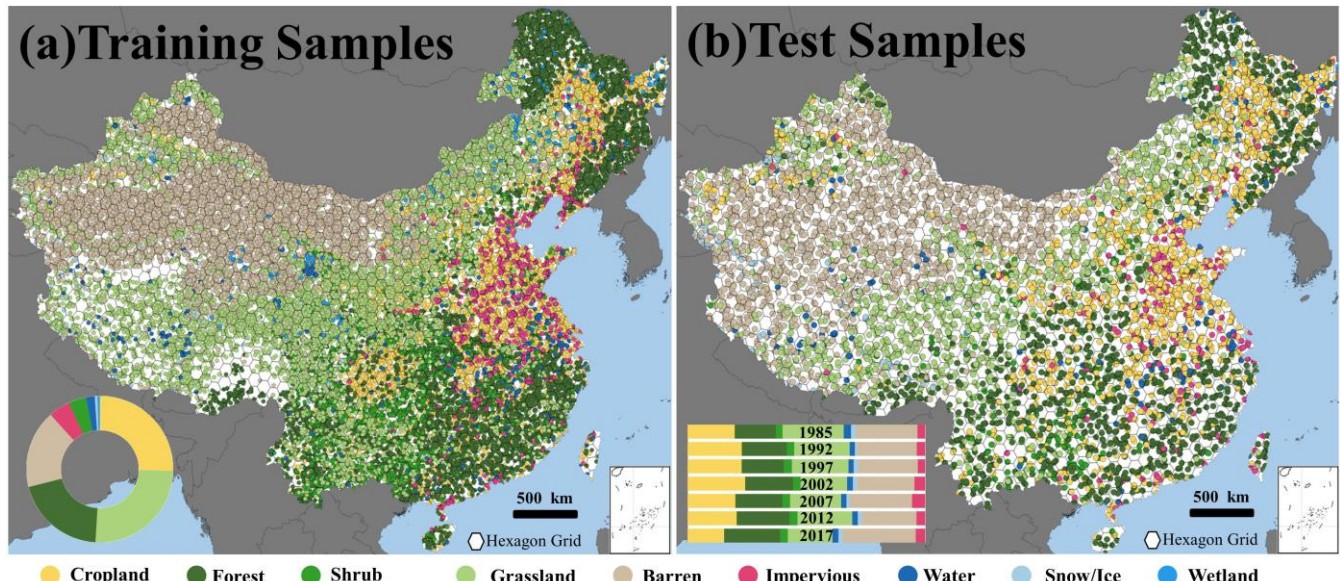


**Figure 2. Spatial distribution of visual-interpreted training and test samples. The proportions of each land cover class were shown in the inner graphs.**

Gong et al., (2019) has demonstrated that it is possible to use training samples of circa 2015 to classify the LC map of 2017. However, given that CLUD was not available after 2015, we manually interpreted 2200 unchanged sites over the entire study

period (i.e., 1985–2019) to further ensure the accuracy of the long time-series products. For manual interpretation, we referred to Google Earth high-resolution images, MODIS EVI and NDVI time-series, Landsat images and its NDVI time-series. Specifically, we first checked the Landsat NDVI time-series (1985–2019) and the MODIS EVI/NDVI time-series (2001–2019) for a candidate sample site. If its NDVI time-series curves were stable (Fig. 3a), the site was regarded as unchanged, and its LC label was then determined via Google Earth images and Landsat images. In particular, for the sites

where Google Map photos or photo spheres were available, these photos were also used to interpret the LC labels. Taking the red dot in Fig. 3 for instance, its NDVI time-series features were stable (Fig. 3a), signifying that it was unchanged and was hence regarded as a potential sample site. It is difficult to determine whether it is bare soil or withered grassland even in high-resolution Google Earth images, owing to its relatively smooth texture. In this case, by courtesy of the Google Map photo sphere, we were able to interpret the actual LC label (i.e., grassland) (Fig. 3b). In total, we selected and interpreted

2200 sites, accounting for 18,000 Landsat pixels. Combined with the aforementioned automatically generated training samples from the CLUD, we finally collected 45,000 training samples in total (Fig. 2a), which were used in the annual classifications.

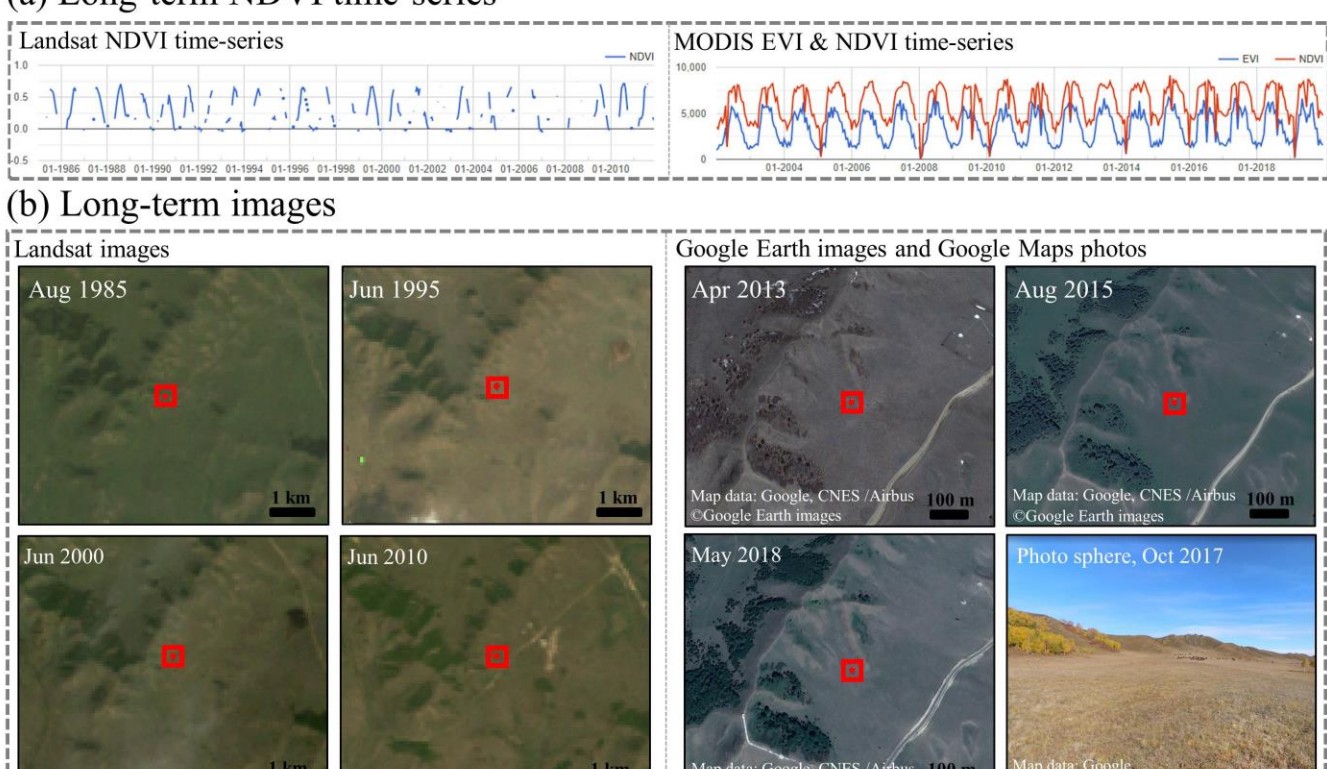

**Figure 3. An example of auxiliary data used to interpret the training samples, including Landsat NDVI time-series, MODIS NDVI and EVI time-series, Landsat images, Google Earth images and Google Maps photos. The red dot represents a sample site located in 42.541067° N, 117.146569° E.**

### 3.4 Classification and spatial-temporal consistency

Random forest (RF) classifier is commonly used for large-scale LC mapping (e.g., Belgiu and Drăgu, 2016; Zhang and Roy, 2017; Zhang et al., 2020) due to a number of advantages, such as the ability to handle high-dimensional data, the tolerance to sample errors and the robustness to missing data (Bauer and Kohavi, 1999; Wulder et al., 2018). Therefore, RF classifier was used to generate the CLCD. The number of trees was set to 200 (Liu et al., 2020a). Based on the training samples, classifiers were trained by input features constructed via Landsat SR from the target year as well as two adjacent years, and the preliminary classification results were obtained by the trained RF classifiers.

To further ensure the accuracy and reliability of the classification results, we proposed a spatial-temporal post-processing method, consisting of spatial-temporal filter and logical reasoning, to refine the time-series mapping results. This method leveraged spatial-temporal context as well as the prior knowledge to suppress the illogical LC conversions. Such errors were usually induced by misclassifications (Wehmann and Liu, 2015). Firstly, the spatial-temporal filter was carried out within a 3×3×3 spatial-temporal window. Specifically, for pixel $i$ with a LC label $L_{i,t}$ in year $t$, if the label of $i$ in pervious year (i.e.,



$L_{i,t-1}$) was not equal to $L_{i,t}$, a LC conversion may take place. In this situation, we further checked the spatial-temporal consistency probability $P_{i,t}$ of pixel $i$ by the following equation:

$$P_{i,t} = \frac{1}{N}[\sum_t^{t+2} \sum_{x-1}^{x+1} \sum_{y-1}^{y+1} I(L_{i,t} = L_j)] \tag{1}$$

where $L_j$ denotes the LC label for pixels in the current window and $N$ represents the total number of pixels (i.e., $N = 27$). $I$ $(L_{i,t} = L_j)$ is the indicator function, i.e., $I = 1$ if $L_{i,t}$ equals to $L_j$ and $I = 0$ otherwise. Besides, $x$ and $y$ indicate the location of $i$. A higher $P_{i,t}$ value signifies the LC conversion, but a lower $P_{i,t}$ value may correspond to a classification error. Here we set a simple rule to check whether the LC change takes place: If the value of $P_{i,t}$ is greater than 0.5 and $L_{i,t}$ is consistent with the dominant LC in the current window, the label of the pixel in year $t$ is considered as changed. A value of $P_{i,t}$ less than 0.5 corresponds to an incorrect classification and hence, the label of this pixel is actually not changed. In such case, $L_{i,t}$ should be corrected as $L_{i,t-1}$. In addition, the CLCD followed the assumption that a LC change should last for more than 2 years (Defourny et al., 2017).

On the other hand, we defined a logical reasoning method via a transition matrix (Table S2) to suppress illogical LC conversions. For example, it is not likely for a pixel to change from barren to cropland within a year. The matrix was inspired by He et al., (2017), but was modified according to the LC changes in China over the past four decades. For instance, China has built many reservoirs in the past 30 years (Li et al., 2018), leading to a mass of cropland covered by water. Thus, conversion from cropland to water should be considered.

### 3.5 Accuracy assessment

In order to assess the accuracy of CLCD, three independent test sets covering the whole China were used: (1) two third-party test sets (i.e. Geo-Wiki and GLCVSS); (2) a visually-interpreted test set (5,463 in total) for seven years (i.e., 2017, 2012, 2007, 2002, 1997, 1992, 1985), with each year containing around 750 samples. To ensure their random distribution, the visually-interpreted points were sampled following the same sampling strategy as the training sample selection (Fig. 2b). Likewise, the Google earth images, Landsat images, and MODIS EVI time-series were used for the interpretation. The spatial distribution of the visually-interpreted test samples was shown in Fig. 2, where the bars represented the proportions of each LC for different years. Finally, the accuracy of CLCD was assessed by confusion matrixes, including Producer's Accuracy (PA), User's Accuracy (UA), Overall Accuracy (OA) and F1-score. The F1-score conveys the balance between PA and UA and is calculated as:

$$F1 = 2\frac{PA \times UA}{(PA + UA)} * 100\% \tag{2}$$

### 3.6 Datasets inter-comparison

In addition to the existing annual LC products (i.e., MCD12Q1 and ESACCI_LC), the CLCD was also compared with several Landsat-derived thematic datasets for more comprehensive quality evaluation. Specifically, with respect to some



dynamic LCs (i.e., impervious surfaces, forests and surface water), we intercompared the CLCD with the Global Forest
Change (GFC) (Hansen et al., 2013), Global Impervious Surface Area (GISA) (available at irsip.whu.edu.cn), Global
Artificial Impervious Area (GAIA) (Gong et al., 2020), Global Annual Urban Dynamics (GAUD) (Liu et al., 2020b) and the
Global Surface Water (GSW) (Pekel et al., 2016). For the GSW dataset, over 3 million Landsat images were used to map
global surface water from 1985 to 2015, with PA more than 95% and UA over 99% (Pekel et al., 2016). The GFC data
depicted global forest changes from 2000 to 2013 using 30 m Landsat date (Hansen et al., 2013). The GISA, GAIA, and
GAUD were Landsat-derived annual ISA (or urban) products for periods 1972–2019, 1985–2018, and 1985–2015,
respectively. Specifically, as suggested by Zhang et al., (2020a), the aforementioned thematic products were aggregated
within the spatial grid of 0.05º×0.05º to obtain the area fraction and the scatterplot and linear regression with the quantitative
metrics of correlation coefficient ($R^2$) and root mean square error (RMSE) were used to demonstrate their agreement.

**Figure 4. Annual China Land Cover Dataset (CLCD) for 1985, 1999, 2009 and 2019.**





## 4 Results and discussion

### 4.1 Accuracy assessment of CLCD

Based on all available Landsat SR data on the GEE, we generated the annual China Land Cover Dataset (CLCD). The accuracy of CLCD was first assessed via visually-interpreted independent samples (Table S3-S9). Overall, the accuracy of
255 CLCD was stable and satisfactory (76.45% < OA < 82.51%), with average OA of 79.30% ± 1.99% (Fig. 5i). For each category, water achieved the highest average F1-score (87.06% ± 7.07%), followed by forest (85.49% ± 1.30%), snow/ice (83.51% ± 7.99%) and barren (81.85% ± 4.15%). The accuracy was relatively high for grassland and impervious area, with mean F1-score over 72%. In addition, CLCD outperformed MCD12Q1 and ESACCI_LC in terms of OA in all the years (Table S10-S19). For the dominant LCs, such as cropland, forest and grassland, CLCD also exhibited better and more stable
F1-scores with respect to the MCD12Q1 and ESACCI_LC (Fig. 5).

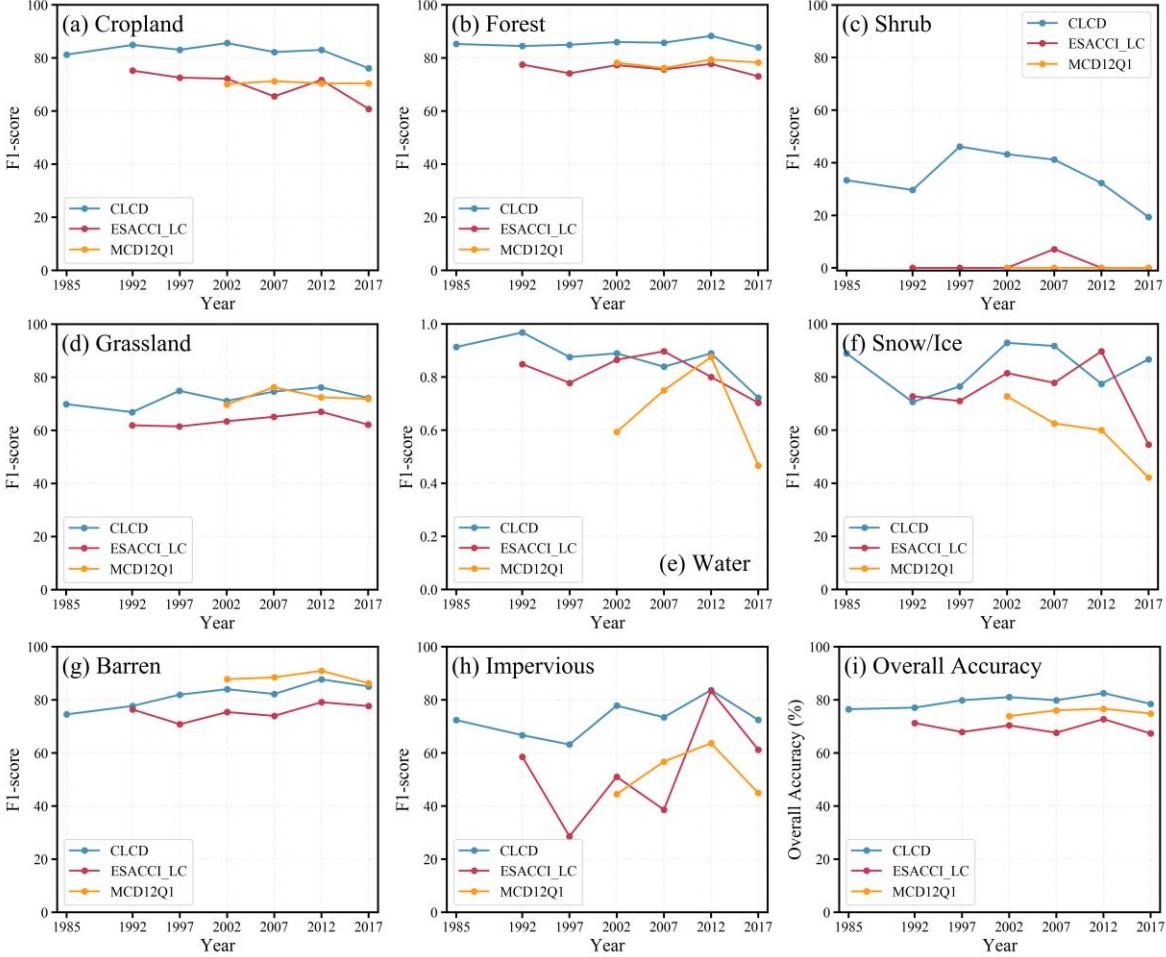

**Figure 5. F1-score and overall accuracy for ESACCI_LC, MCD12Q1 and CLCD based on visual-interpreted test samples.**





**Table 2. Comparison of mapping accuracy based on Geo-Wiki test samples for EASCCI_LC, MCD12Q1, FROM-GLC and CLCD*.**

| | | Geo-Wiki | | | | | | | | OA (%) |
|---|---|---|---|---|---|---|---|---|---|---|
| | | Cropland | Forest | Shrub | Grassland | Water | Snow/Ice | Barren | Impervious | |
| CLCD | PA (%) | 73.66 | 72.68 | 43.48 | 15.95 | 37.84 | 80.95 | 56.96 | 48.96 | |
| | UA (%) | 77.73 | 85.24 | 2.82 | 48.28 | 63.64 | 18.89 | 8.91 | 66.20 | 54.57 |
| | F1 (%) | 75.64 | 78.46 | 5.30 | 23.98 | 47.46 | 30.63 | 15.41 | 56.29 | |
| MCD12Q1 | PA (%) | 79.16 | 67.58 | 16.67 | 15.37 | 50.00 | 100 | 52.34 | 57.89 | |
| | UA (%) | 64.56 | 87.22 | 0.28 | 51.72 | 36.36 | 7.78 | 13.27 | 61.97 | 51.97 |
| | F1 (%) | 71.12 | 76.15 | 0.55 | 23.70 | 42.10 | 14.44 | 21.17 | 59.86 | |
| EASCCI_LC | PA (%) | 54.66 | 79.07 | 0.00 | 15.54 | 53.57 | 85.71 | 61.54 | 75.00 | |
| | UA (%) | 86.56 | 69.05 | 0.00 | 43.79 | 68.18 | 13.33 | 7.92 | 67.61 | 50.87 |
| | F1(%) | 67.01 | 73.72 | 0.00 | 22.94 | 60.00 | 23.07 | 14.03 | 71.11 | |
| FROM-GLC | PA (%) | 73.56 | 64.46 | 13.89 | 14.40 | 23.21 | 77.78 | 62.41 | 62.50 | |
| | UA (%) | 48.75 | 81.50 | 1.41 | 45.17 | 59.09 | 7.78 | 34.85 | 49.30 | 49.23 |
| | F1 (%) | 58.64 | 71.99 | 2.56 | 21.84 | 33.33 | 14.15 | 44.73 | 55.12 | |

*PA, UA and OA are abbreviations for the producer's accuracy, user's accuracy and overall accuracy respectively. The F1 represents the harmonic mean of the PA and the UA.

To better validate the accuracy of CLCD, we used the Geo-Wiki samples to intercompare the CLCD, FROM-GLC, MCD12Q1, and ESACCI_LC. FROM-GLC was the first global 30 m LC data (Gong et al., 2013). Here we adopted its second generation product, which was generated using Landsat images acquired from 2013 to 2015 (Li et al., 2017). Overall,

the CLCD possessed a better OA of 54.57% against the ESACCI_LC (50.87%), the MCD12Q1 (51.97%) and the FROM-GLC (49.23%), respectively (Table S20-S23). Specifically, CLCD achieved better accuracy than FROM-GLC in most LCs and showed similar performance for the impervious area (Table 2).

We also compared the accuracy of different products (i.e., CLCD, GlobaLand30, MCD12Q1 and ESACCI_LC) using the GLCVSS sample set. In particular, GlobaLand30 was also included for comparison to our CLCD. GlobaLand30 was a 30 m

global LC data for 2001 and 2010, produced using a pixel-object-knowledge approach (Chen et al., 2015). We used GlobaLand30 in the year of 2010 for the comparison. It was found that the CLCD obtained the highest accuracy of 65.64%, outperforming the ESACCI_LC (57.16%), the MCD12Q1 (61.66%), and the GlobaLand30 (63.12%), respectively (Table S24-S27). Although the UA or PA of CLCD did not always outperform that of other products, CLCD achieved the highest F1-scores in nearly all the LCs (Table 3).

In summary, CLCD achieved higher OA with respect to the existing LC products (i.e., MCD12Q1, ESACCI_LC, FROM-GLC, and GlobaLand30), based on the visually-interpreted and third-party samples. In addition, the temporal coverage of CLCD spans 35 years (1985–2019), which exceeds the ESACCI_LC (1992–2018) and the MCD12Q1 (2001–2018).

**Table 3. Comparison of mapping accuracy based on GLCVSS (Global Land Cover Validation Sample Set) test samples for EASCCI_LC, MCD12Q1, GlobaLand30 and CLCD\*.**

| | | GLCVSS | | | | | | | | |
| | | Cropland | Forest | Shrub | Grassland | Water | Snow/Ice | Barren | Impervious | OA (%) |
| --- | --- | --- | --- | --- | --- | --- | --- | --- | --- | --- |
| CLCD | PA (%) | 64.52 | 79.47 | 0.00 | 33.83 | 72.73 | 94.44 | 91.26 | 64.29 | |
| | UA (%) | 78.43 | 88.65 | 0.00 | 72.70 | 64.00 | 21.25 | 53.29 | 52.94 | 65.46 |
| | F1 (%) | 70.80 | 83.81 | 0.00 | 46.17 | 68.09 | 34.69 | 67.29 | 58.07 | |
| MCD12Q1 | PA (%) | 61.76 | 69.74 | 0.00 | 32.26 | 88.89 | 72.73 | 88.93 | 45.45 | |
| | UA (%) | 58.82 | 84.34 | 0.00 | 70.92 | 32.00 | 10.00 | 59.33 | 29.41 | 61.66 |
| | F1 (%) | 60.25 | 76.35 | 0.00 | 44.35 | 47.06 | 17.58 | 71.18 | 35.71 | |
| EASCCI_LC | PA (%) | 46.82 | 81.54 | 18.18 | 28.36 | 80.00 | 85.71 | 92.76 | 71.43 | |
| | UA (%) | 84.59 | 68.30 | 2.50 | 62.06 | 48.00 | 15.00 | 46.44 | 39.22 | 57.16 |
| | F1(%) | 60.28 | 74.34 | 4.40 | 38.93 | 60.00 | 25.53 | 61.89 | 50.64 | |
| GlobaLand30 | PA (%) | 62.33 | 79.80 | 15.38 | 32.67 | 100 | 75.00 | 89.71 | 62.79 | |
| | UA (%) | 79.27 | 77.30 | 2.50 | 75.18 | 40.00 | 18.75 | 53.83 | 52.94 | 63.12 |
| | F1 (%) | 69.79 | 78.53 | 4.30 | 45.55 | 57.14 | 30.00 | 67.29 | 57.45 | |

\*PA, UA and OA are abbreviations for the producer's accuracy, user's accuracy and overall accuracy respectively. The F1 represents the harmonic mean of the PA and UA.

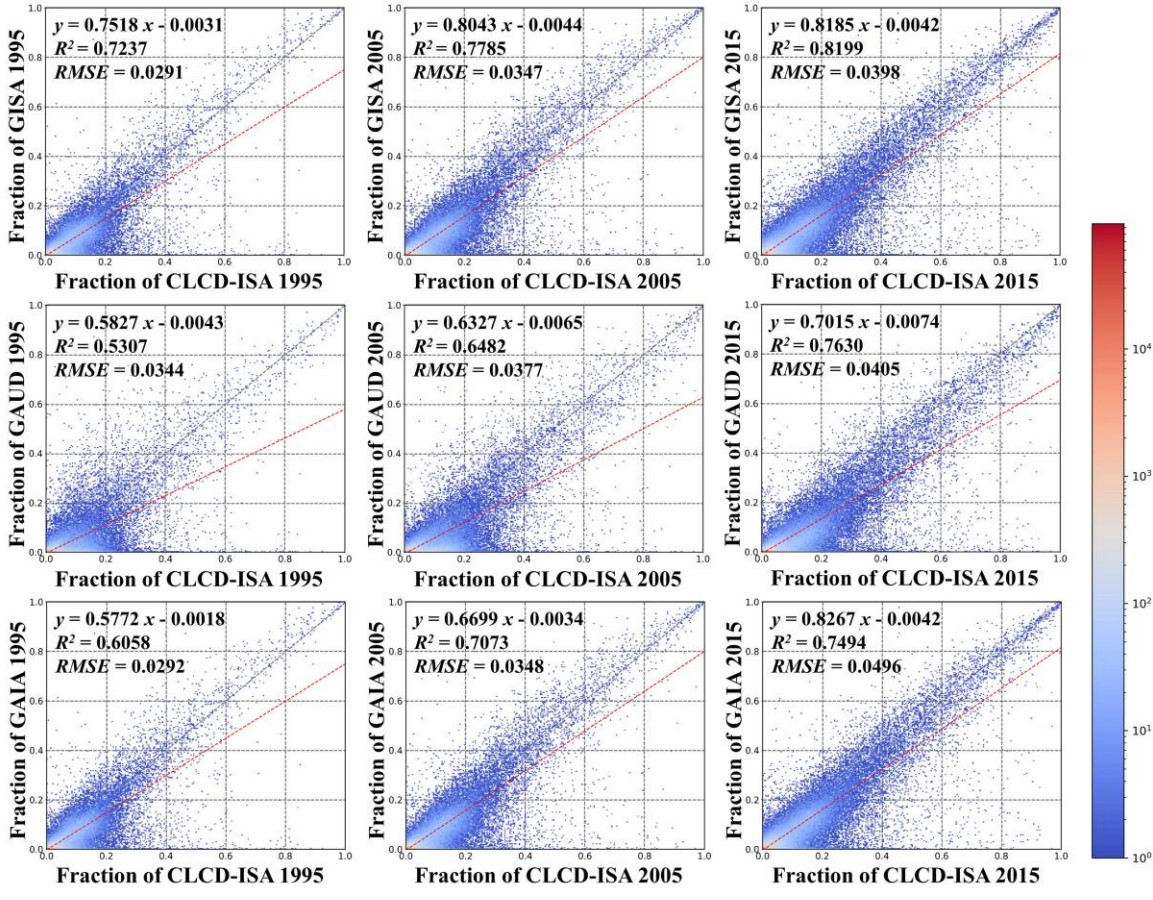

**Figure 6. Scatterplots of ISA fraction between different products in 1995, 2005 and 2015, respectively. ISA fraction was aggregated within the 0.05° by 0.05° spatial grid.**



## 4.2 Inter-comparison with existing 30 m thematic products

### 4.2.1 Comparison with ISA

The impervious Surface Area (ISA), as the consequence of urbanization, has rapidly sprawled in the past few decades and had significant implications on regional ecological changes (Goldewijk, 2001). In view of the fast urbanization in China, the accuracy of CLCD can be validated by the accuracy of ISA. Thereby, we compared the ISA of CLCD (CLCD-ISA) with the existing well-known 30 m annual ISA products (i.e., GISA, GAUD, GAIA). Without loss of generality, the ISA products were selected at a 10-year interval for three periods (i.e., 1995, 2005, and 2015), and the ISA fractions were calculated within the 0.05°×0.05° spatial grid. Overall, the CLCD-ISA showed good consistency with the existing ISA products (0.53 < $R^2$ < 0.82), indicating that the CLCD products are reliable (Fig. 6). Although good agreement has been found between CLCD-ISA and other products in most years, the correlation between CLCD-ISA and GAUD in 1995 was only 0.53. This was probably subject to the underestimations of villages in GAUD during early years since GAUD focused on the urban areas. It can be seen in Fig. 7 that CLCD-ISA and GISA were generally similar while GAIA and GAUD had a little omission over the North China Plain where villages gather.

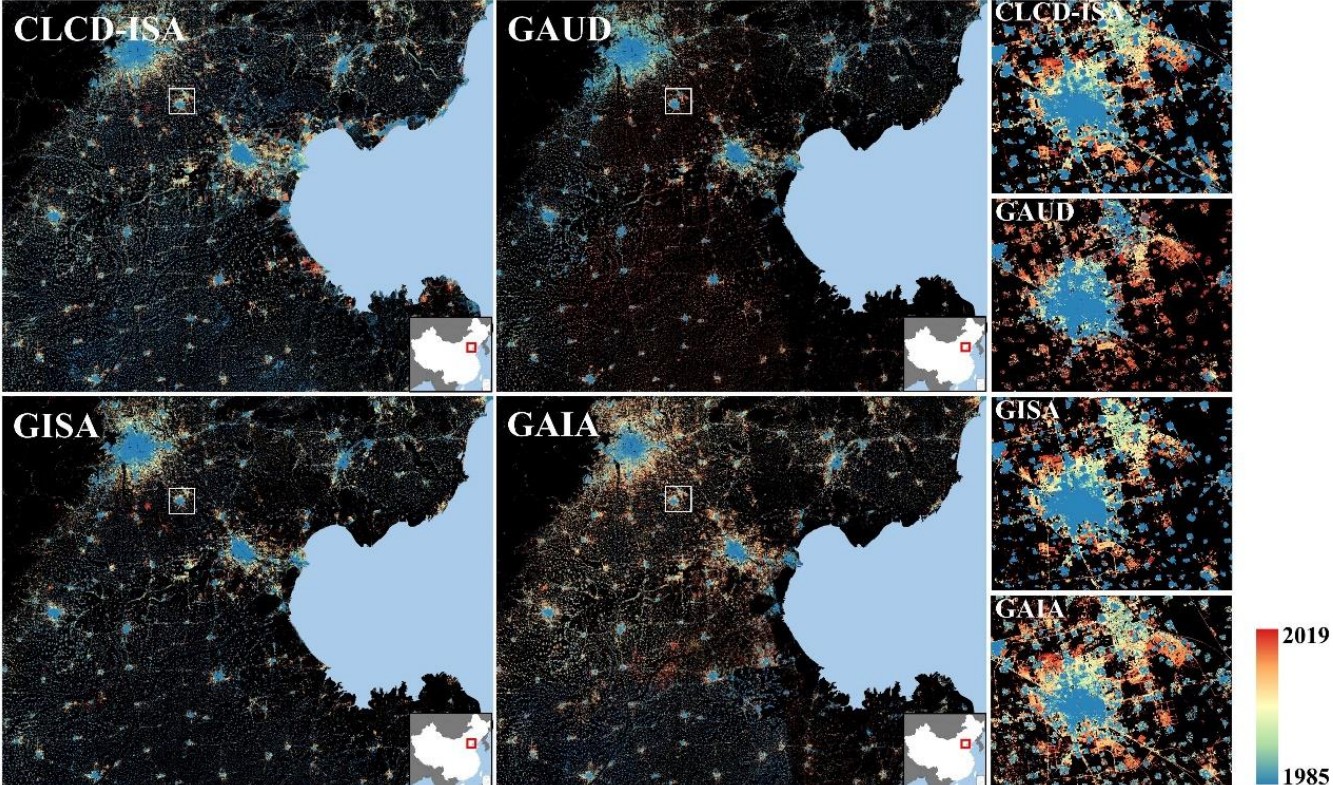

**Figure 7. Comparison between different ISA products over North China Plain, with zoom-in maps located in Langfang City (39.526545° N, 116.703692° E).**

Earth System Science Data
en




### 4.2.1 Comparison with forest change

We further compared the foreste in CLCD (CLCD-forest) with the Global Forest Change (GFC) data from Hansen et al.,(2013) to demonstrate the accuracy of CLCD. The GFC data (v1.7) included: forest cover as fraction (2000), forest gain

(2001–2019 as total) and year of forest loss. Since the year of forest gain was unavaliable, we selected the areas with forest cover greater than 30% as forest to obtain the 2000 and 2019 forest map, as suggested by Taubert et al., (2018). Based on the above forest maps, the forest fraction was aggregated within the 0.05°×0.05° spatial grid. It was found that CLCD-forest showed relatively high agreement with the GFC ($0.84 < R^2 < 0.87$), signifiying the reliability of CLCD. Additionally, as can be seen in the Fig. 9, the spatial distribution of CLCD-forest was gengerally similar to the GFC data.

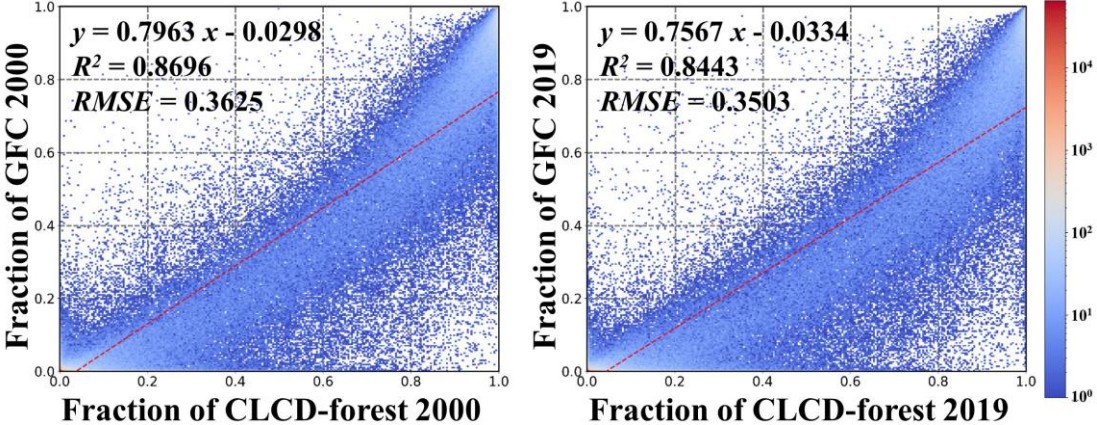

315
**Figure 8. Scatterplots of forest fraction between the CLCD and GFC in 2000 and 2019. Forest fraction was aggregated within the 0.05° by 0.05° spatial grid.**

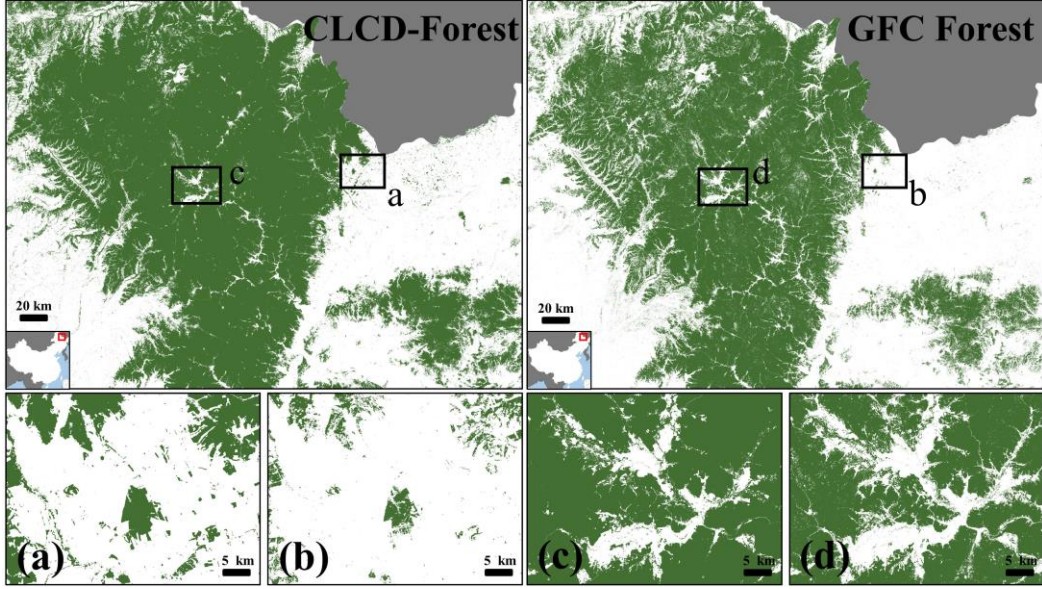

**Figure 9. Example of 2019 forest maps for CLCD and GFC located at Lesser Khingan Mountains (47.703792° N, 129.388820° E).**



### 4.2.1 Comparison with surface water

Surface water in China has changed dramatically over the past decades due to comprehensive implications of human activities and climate changes (Lutz et al., 2014; Yang et al., 2020a). Therefore we assessed the quality of the CLCD dataset through inter-comparsion of the Global Surface Water (GSW) (Pekel et al., 2016). The GSW (v1.2) data were the first 30 m dataset that documented the monthly persistence and existence of surfaec water from 1985 to 2018. It should be noted that CLCD used the annual median reflectance (i.e., 50th percentile), and hence, its water extent (CLCD-water) was close to the average annual water extent. The GSW, on the other hand, had a denser temporal sequence (monthly). Accordingly, to facilitate the inter-comparison, we obtained GSW average annual water extent based on the intra-annual water occurrence (Yang et al., 2020a). In this manner, we selected GSW and CLCD-water in 1995, 2005 and 2015, and calculated the water fraction of the two datasets within the $0.05° \times 0.05°$ spatial grid. As demonstrated in Fig. 10, the high consistency ($0.86 < R^2 < 0.96$) was achieved with two products, indicating the reliability of CLCD.

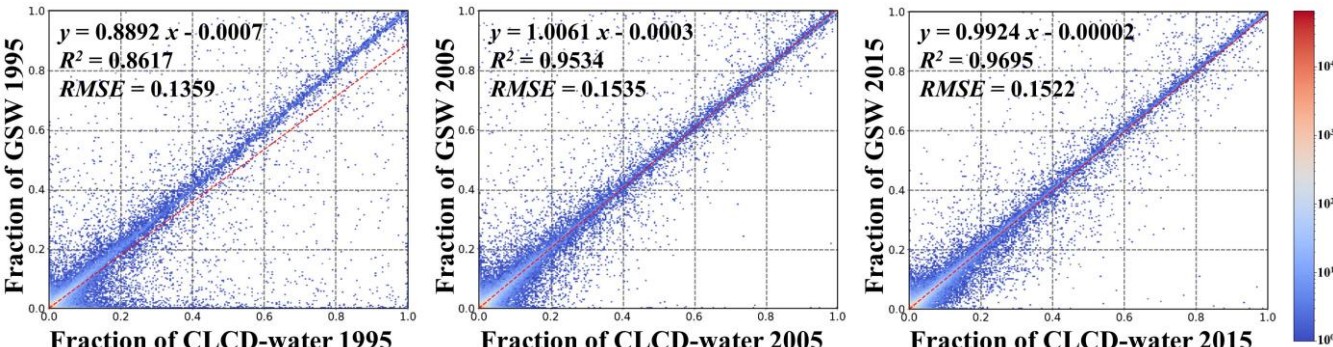

**Figure 10. Scatterplots of surface water fraction between CLCD and GSW in 1995, 2005 and 2015, respectively. Water fraction was aggregated within the 0.05° by 0.05° spatial grid.**

To better explain the difference of two products in depicting water dynamics, we counted the number of water occurrences from 1985 to 2018. A higher occurrence signifies permanent water while a lower occurrence indicates seasonal or new permanent water. As shown in Fig. 11, the CLCD-water extent was closely similar to the GSW water extent, which again demonstrated the reliability of CLCD. However, we noted that GSW captured more seasonal water (Fig 11b & 11d). This was expected, as GSW had a denser temporal sequence (monthly) than CLCD (annual). Consequently, it is difficult for CLCD to capture short-term fluctuations (e.g., flooding). However, long-term changes caused by anthropogenic activities, such as reservoir construction, were accurately observed by CLCD (Fig. 11a & 11c).

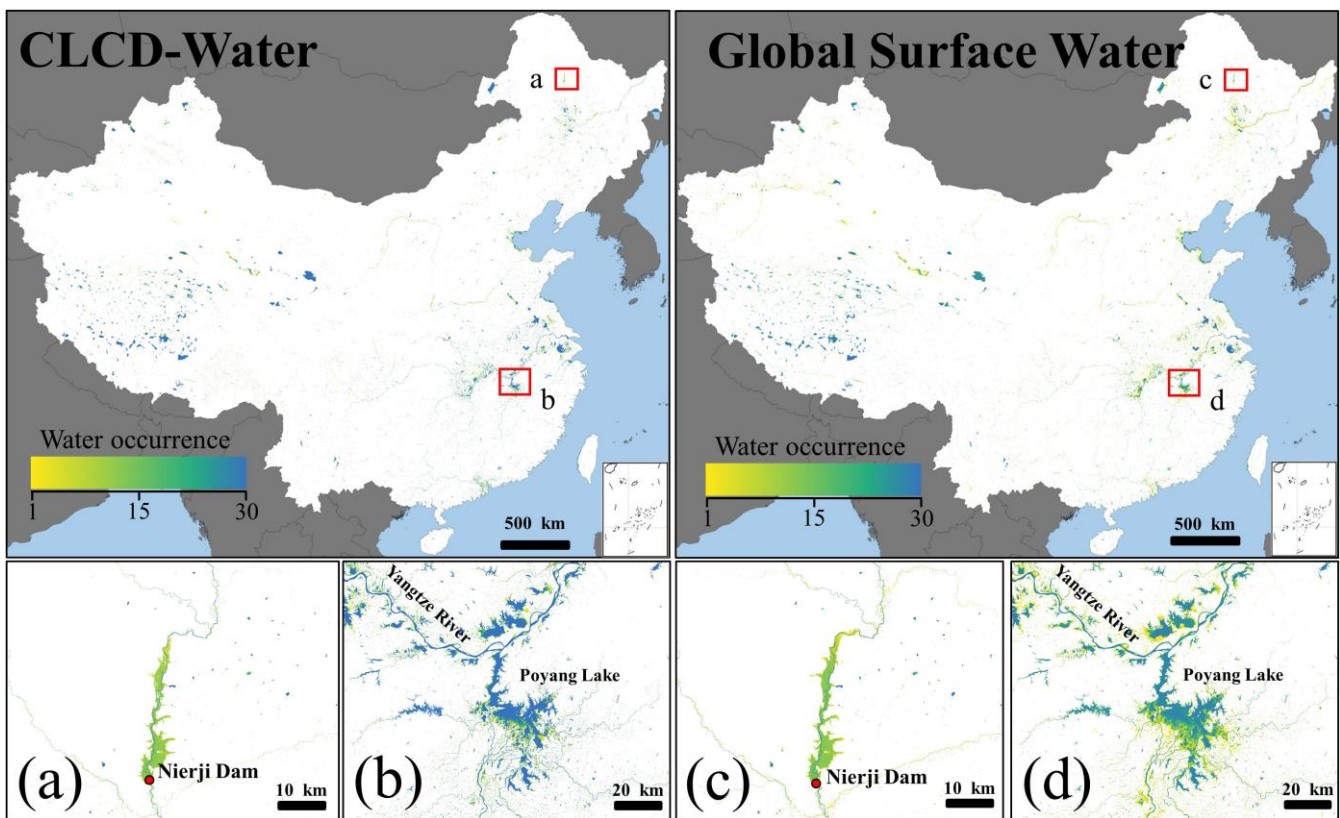

**Figure 11.** Annual water occurrence (1985–2018) derived from water extent of CLCD and GSW, with zoom-in maps located in Nierji Dam (a & c) and Poyang Lake (b & d).

## 4.3 LC dynamics in China 1985–2019

### 4.3.1 General temporal trend

Based on the CLCD generated in this study, we analysed the LC changes in China from 1985–2019 (Fig. 12). The impervious area has unprecedentedly sprawled over the past 35 years, with more than 24.5 million ha in 2019, which was increased by 1.5 times relatively to that in 1985. In terms of change magnitude, the impervious area also exceeded the rest, with 46.56% more than the second-ranked forest. The area of surface water increased by 2.37 million ha, 78.40% of which occurred after 1995 when the development of hydropower was proposed by the Ninth Five-Year Plan of China (Li et al., 2018). The increasing reservoirs resulting from dam construction are one of the reasons accounted for the surface water extension (Yang et al., 2020a). Although extensive reclamation has undergone in the northeast and northwest China to feed the growing population, cropland was generally decreased. In particular, 4.57% of cropland was lost during 1985–2010, but it should be emphasized that only 0.03% was lost after the implementation of Red Lines of Cropland in 2010 (Xie et al., 2018). Due to a series of afforestation policies in China, such as the Three North Shelterbelt project started in the 1980s and the Gain for Green project initialized after 2000, the forest had increased by 4.34% (10.02 million ha) from 1985 to 2019



(Fig. 12b). The shrub decreased by 2.59 million ha, with similar decrease trends found with ESACCI_LC (Fig. 12c) and (Liu et al., 2020a). The barren increased slightly by 0.80% from 1985 to 2000, but decreased by 2.62% from 2000 to 2019. The decrease of the barren may be related to the ecological project of Returning Grazing Land to Grassland after 2003 (Xiong et al., 2016). In contrast, grassland decreased by 2.15% (6.23 million ha) before 2000, but only 1.16% (3.28 million ha) from 2000 to 2019 thanks to the implementation of grassland conservation policies such as the Gain For Green after 2000 (Li et al., 2017). The wetland decreased by 0.92 million ha, 91.3% of which occurred before 2000, which was likely attributed to the extensive reclamation occurred in the Sanjiang Plain of Northeast China before 2000 (Zhang et al., 2010, 2009).

Compared with ESACCI_LC and MCD12Q1, CLCD showed generally similar trends. However, it should be noticed that the CLCD detected more surface water and impervious area with respect to two coarse-resolution LC products (Fig. 12). This was attributed to the discernible resolution (30 m) of Landsat data, which allow better delineation of relatively small LC patches. Therefore, the 30 m CLCD data are more applicable in the fine-scale environment and land surface processes simulation.

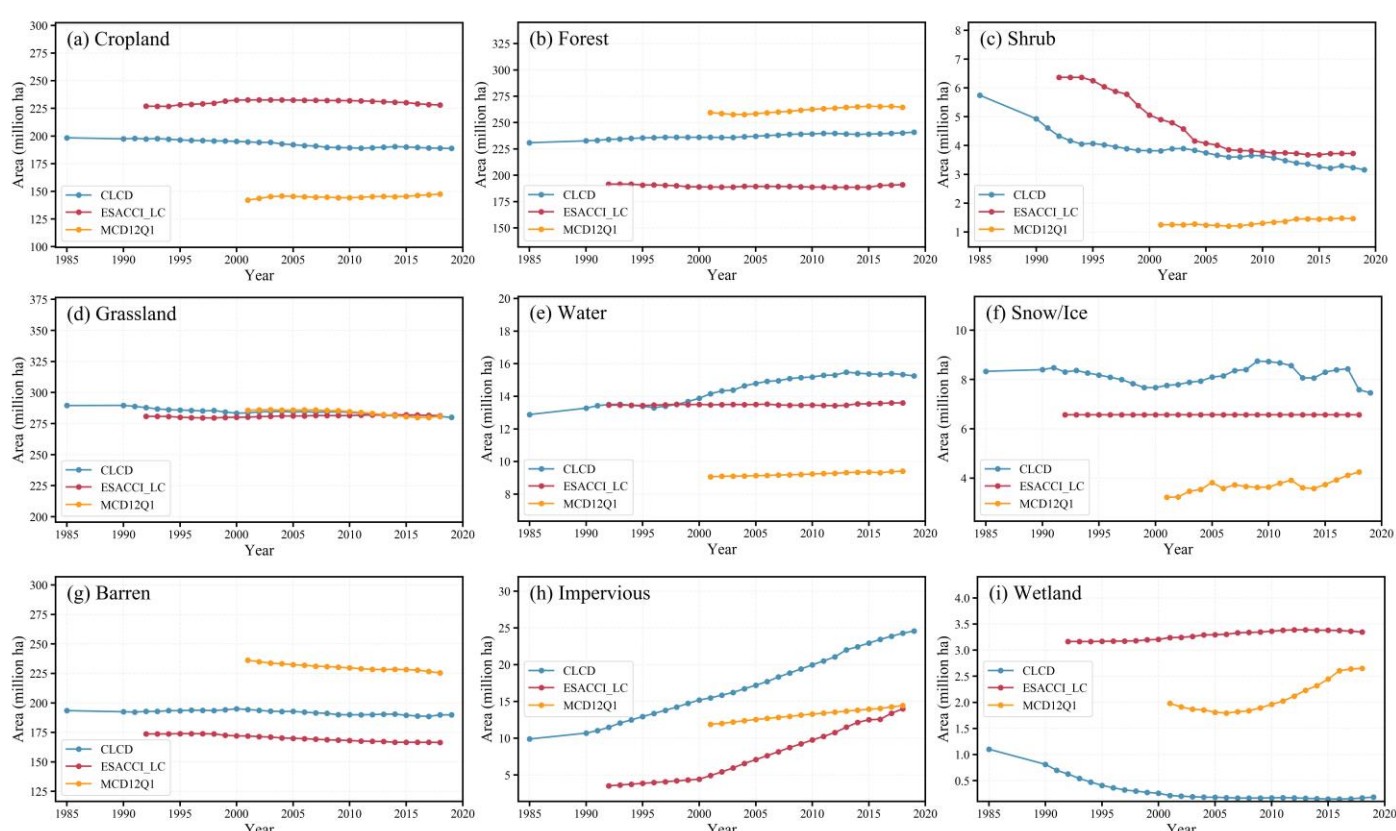

**Figure 12. Temporal changes in area of different land covers in China from 1985 to 2019.**



### 4.3.2 LC conversion patterns

In addition to depicting the temporal trends, CLCD data can further reveal LC conversions, which are also important to global change studies. Therefore, we analysed the major LC conversions in China from 1985–2019 (Table 5). Overall, cropland loss accounted for the highest proportion among all conversions (33.50%), followed by the grassland loss (29.55%).

The main conversion directions of cropland were the impervious area (32.03%) and forest (35.07%), reflecting the unprecedented urbanization process in China (Fig. 13b & 13i) and the Gain for Green project (Fig. 13h), respectively. Grassland (38.35%) and forest (46.79%) were the main sources converted to cropland (Table 4), indicating the reclamation in the northeast (Fig. 13c) and northwest China (Fig. 13g & 13e), respectively. Meanwhile, the interconversion between barren and grassland was also found (Liu et al., 2020a). Additionally, afforestation was also a major cause (25.72%) of

grassland loss. 84.36% of the impervious area gain came from cropland, which can be seen in Fig. 4 (e.g., North China Plain and Eastern China). 17.3% of the water gain stemmed from the barren and grassland (e.g., Fig. 13a), which were closely associated with the rapid lake expansion in the Tibetan Plateau. The accelerated glacier melts and increased precipitation expanded those lakes (Lutz et al., 2014; Song et al., 2014). It was noteworthy that about 4% of the impervious area was originated from water (Fig. 13a), while 48.11% of the water loss was induced by the reclamation, which has been a common

phenomenon in the middle and lower Yangtze drainage region (Du et al., 2011). Lake shrinkage caused by the reclamation and urban sprawl in such regions has triggered some problems concerning the water resource management and flood relief (Hou et al., 2020; Xie et al., 2017). In particular, Figure 13j demonstrated the conversion of the barren to grassland in the Mu Us Desert of Yulin City, where green area has increased significantly over the past few decades due to several ecological restoration projects implemented by the government (Wang et al., 2020; Xiu et al., 2018). Overall, the CLCD shows great

potential to reveal the human impact on LC changes under the condition of climate change, and also demonstrates the promising applications in environment change studies.

**Table 4. Ratio (%) of land cover conversions from 1985 to 2019. The blue colour denotes a higher ratio and the yellow colour represents a lower ratio.**

| | | 2019 | | | | | | | | |
|---|---|---|---|---|---|---|---|---|---|---|
| | | Cropland | Forest | Shrub | Grassland | Water | Snow | Barren | Impervious | Wetland |
| | Cropland | - | 11.75 | 0.25 | 8.97 | 1.66 | 0 | 0.14 | 10.73 | 0 |
| | Forest | 11.87 | - | 0.82 | 0.38 | 0.06 | 0 | 0 | 0.42 | 0 |
| | Shrub | 0.57 | 2.47 | - | 0.54 | 0 | 0 | 0 | 0 | 0 |
| | Grassland | 9.73 | 7.6 | 0.33 | - | 0.78 | 0.21 | 10.22 | 0.68 | 0.04 |
| 1985 | Water | 0.89 | 0.11 | 0 | 0.14 | - | 0.01 | 0.18 | 0.52 | 0 |
| | Snow | 0 | 0.02 | 0 | 0.41 | 0.14 | - | 1.33 | 0 | 0 |
| | Barren | 1.58 | 0.01 | 0 | 11.06 | 0.95 | 0.94 | - | 0.36 | 0 |
| | Impervious | 0.06 | 0 | 0 | 0 | 0.23 | 0 | 0.01 | - | 0 |
| | Wetland | 0.67 | 0.07 | 0 | 0.06 | 0.01 | 0 | 0 | 0.01 | - |



**Figure 13. Typical land cover changes and conversions observed in China from 1985 to 2019: (a) lake shrinkage in Wuhan City (30.543703° N, 114.296992° E); (b) cropland loss due to the urbanization in Shanghai City (31.214542° N, 121.490796° E); (c) deforestation induced by reclamation in northeast China (49.508726° N, 124.939498° E); (d) bare land converted to grassland in the Qira oasis, located at the southern fringe of the TAKLAMAKAN Desert (36.237807° N, 81.215592° E). The illustration is of Landsat images with false-colour combination (R: NIR, G: Red, B: Green) to enhance the grassland; (e) grassland loss due to the reclamation in northwest China (43.575843° N, 80.987817° E); (f) Siling Co Lake expansion in the Tibet Plateau (31.793826° N, 89.057216° E); (g) reclamation in the Aksu Prefecture (40.593062° N, 81.048109° E); (h) afforestation in Yunnan Province (26.518645° N, 103.613297° E); (i) Expansion of Chengdu City (30.666955° N, 104.068452° E); (j) grassland gain in the Mu Us Desert, Yulin City (38.509526° N, 109.660052° E).**



## 4.4 limitations and future work

CLCD enables fine-scale annual LC monitoring over China by combining long-term 30 m Landsat archive and cloud-based geospatial analysis platform. One of the major limitations to CLCD is the uneven spatial and temporal coverage of Landsat 5. As the only operational platform prior to 1999, Landsat 5 had no on-board storage and lost its relay capability in 1992 (Wulder et al., 2016). Thus, data transmission was limited to the line-of-sight of the international receiving stations (Loveland and Dwyer, 2012). Courtesy of the Landsat Global Archive Consolidation (LGAC) initiative (Wulder et al., 2016),

old acquisitions resided by these international receiving stations were continuously recovered to an accessible global archive. Moreover, Landsat 5 followed a commercial pre-order acquisition plan before 1990, which further limited its availability before 1990 (Loveland and Dwyer, 2012; Pekel et al., 2016). Therefore, we used all Landsat SR captured before 1990 to generate the CLCD of 1985 to minimize the influence induced by the availability of Landsat 5. As the Landsat archive enriches through the commissioned platforms (i.e., Landsat 7 and 8) and the LGAC, we will be able to extend the temporal

coverage of the CLCD. In addition, the Multispectral Scanner System (MSS) on board Landsat 1-5 provides four spectral bands at 60 m spatial resolution. Thus, future attempts would backdate the LC monitoring to 1970s by incorporating Landsat MSS. On the other hand, there are multiple sources of data from platforms orbiting concurrently with the Landsat satellites. These data can be further employed to update and strengthen the CLCD, such as the Sentinel-2 satellites equipped with red-edge bands (20 m) and the Sentinel-1 satellites (10 m) that measure the dielectric properties and roughness. The overarching

objective of this research, however, is to generate a long-term annual LC dataset for China. To this aim, Landsat images are more appropriate due to their fine spatial resolution and long-term time span.

## 5 Data availability

   The CLCD product generated in this study is available in public domain at http://doi.org/10.5281/zenodo.4417810 (Yang and Huang, 2021). The CLUD datasets were provided by the Resource and Environment Science and Data Center (available

at www.resdc.cn/DOI/doi.aspx?DOIid=54). The Landsat SR, SRTM, GSW(v1.2), GFC(v1.7) and MCD12Q1 were both acquired from the Google Earth Engine (available at code.earthengine.google.com). ESACCI_LC was provided by the European Space Agency climate office (available at climate.esa.int/en/projects/land-cover). Geo-Wiki test samples were obtained from the reference campaign (available at doi.pangaea.de/10.1594/PANGAEA.869680). The GlobaLand30 and GAUD were downloaded from the website of National Geomatics Center of China (available at www.globallandcover.com)

and Sun Yat-sen University (available at doi.org/10.6084/m9.figshare.11513178.v1). The FROM-GLC, GLCVSS, GAIA were assessed from the Tsinghua University (available at data.ess.tsinghua.edu.cn). The GISA was provided by the Institute of Remote Sensing Information Processing in Wuhan University (available at irsip.whu.edu.cn).



## 6 Conclusion

LC is a fundamental parameter for environment and climate change studies. Rapid economic and population growth in China
over the past few decades has tremendously altered its land cover (LC). Therefore, sequential and fine-resolution LC
monitoring of China is important to implement informed and sustainable management. While the LC monitoring via
satellites is increasingly recognized, fine-resolution annual LC and its dynamics in China via the remote sensing approach
have rarely been investigated in the current literature. Therefore, to better understand the LC changes in China, we generated
the first Landsat-derived annual China Land Cover Dataset (CLCD) from 1990–2019 based on the GEE platform. The
CLCD has higher spatial resolution and longer temporal coverage with regard to the existing annual LC products (i.e.,
MCD12Q1 and ESACCI_LC). The overall accuracy of CLCD reached 79.31% based on 5,463 independent test samples. In
addition, assessment using 5,131 third-party validation samples showed that the overall accuracy of CLCD exceeded that of
MCD12Q1, ESACCI_LC, FROM_GLC, and GlobaLand30. The accuracy and reliability of CLCD was further validated by
comparison with Landsat-derived thematic datasets. The CLCD-based dynamic analysis revealed temporal trends and
patterns of LC conversions in China, such as expansion of impervious surface and water, cropland reduction, forest
increment, and grassland loss. We also found several significant LC conversions, such as the conversion of cropland to forest
and the impervious, barren loss induced by grassland gain, grassland loss caused by the reclamation and afforestation, and
water loss resulted from cropland sprawl, signifying the rapid urbanization as well as a series of ecological projects in China.
Annual LC information is important for environmental and climate change studies. The CLCD provides a fine-scale view of
LC and its long-term changes in China at 30 m resolution. As there are increasing environment and climate modelling
studies using annual LC dataset, CLCD can provide important reference for national or regional modelling studies. The
CLCD, combined with other data (e.g., water cycle data), will allow for more comprehensive characterization of
environment and climate changes.

**Author contributions.** Conceptualization: Xin Huang; Investigation: Jie Yang; Methodology: Xin Huang and Jie Yang;
Software: Jie Yang; Validation, Jie Yang; Writing – original draft preparation: Xin Huang and Jie Yang; writing – review
and editing: Xin Huang and Jie Yang.

**Competing interests.** The authors declare that they have no conflict of interest.

**Financial support.** The research was supported by the National Natural Science Foundation of China under Grants
41771360 and 41971295, and the National Program for Support of Top-notch Young Professionals.

**Acknowledgments.** The authors greatly appreciate the free access of the Landsat data provided by the USGS, the
ESACCI_LC product provided by the European Space Agency, the MCD12Q1 product provided by the National



Aeronautics and Space Administration, the GlobaLand30 provided by the National Geomatics Center of China, the GSW data provided by the Joint Research Center of European Union and Google, the GFC data provided by the University of Maryland and Google, the test samples provided by the Geo-Wiki, the GAIA, GLCVSS and FROM-GLC provided by Tsinghua University, the GAUD data provided by Sun Yat-sen University, and the GISA data provided by Wuhan

University. Special thanks to the Google Earth Engine team for their excellent work to maintain the planetary-scale geospatial cloud platform.

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
