# Peer review of "m annual land cover and its dynamics in China from 1990 to 2019"

_Earth System Science Data, 2021_

## Author Comment (AC1)

Referee #1

This is an interesting and well-written paper. The authors seek to develop an annual land cover dataset of China, and such kind of continuous time-series products are lacking and urgently needed in the current research. In this context, the authors first derived training samples from existing data and visual-interpretation, and then fed it to random forest classifier to obtain land cover results. They compared the results against those from the MODIS, CCI and GlobaLand30, with a relatively better accuracy achieved. In particular, it is very interesting to see the detailed comparison between the CLCD datasets and other well-known thematic products (e.g., GISA, GAIA, GFC, GSW). In addition, the authors also examined spatiotemporal patterns of land cover changes.

The study is technically sound. The use of the Landsat and Google Earth Engine for 30 m land cover mapping is a good choice, especially given the post-processing to ensure the consistency. The validation of the result is generally comprehensive and reliable. The analysis in the discussion section flows reasonably from the results. As such, I recommend that it be published after minor revisions.

1) Line 22. In the title, the CLCD annual dataset spans 1990-2019. But, the land cover changes spans 1985-2019. Can you clarify this issue?

**R:** Thanks for your comment. Due to the missing data of Landsat satellites during 1985 to 1990, the CLCD was generated annually after the 1990. But we generated a map for 1985 as a supplement using all available data before 1990. Therefore, we claimed that the CLCD annual dataset spans 1990-2019 while the land cover changes spans 1985-2019.

2) Line 52. It is vague to use "over 90,000 visually-interpreted training samples". Please explicitly identify how many samples were used.

**R:** Thanks for your comment. The corresponding sentence has been revised as:

" *91,433 visually-interpreted training samples*".

3) Line 91. What do you mean by "inconsistency between different Landsat sensors"?

R: Many thanks. The inconsistency refers to the different band widths between different Landsat sensors. We therefore revised the sentence as:

"*inconsistent band widths between different Landsat sensors*"

4) Line 111. Why only "Quite Sure" Geo-Wiki samples were chosen?

R: Thanks for pointing out this issue. There are four confidence levels (i.e., unsure, less sure, sure and quite sure) for Geo-Wiki samples (Fritz et al., 2017). We only selected the samples with the two highest confidence levels (i.e., quite sure and sure) for validation to ensure the reliability of the accuracy assessment. According to your comment, we revised the corresponding sentence as:

"*Based on the quality flag, we selected 3,000 "Quite Sure" and "Sure" Geo-Wiki samples located in China*"

5) Line 173. "trainings" should be "training".

R: Corrected.

6) Line 216. I'm not sure what do you mean by saying "dominant LC" here. Please clarify.

R: Thanks very much. The "dominant LC" refers to the land cover accounted for more than half of the total pixels (i.e., $P_{i,t} > 0.5$) in the current spatial-temporal window. We have revised the corresponding description in the revised manuscript.

7) Line 237. It is interesting to compare against these existing datasets. But please clarify the reason why MODIS and CCI were chosen, since they have relatively coarse resolution.

R: Thanks for your comment. We inter-compared the CLCD with MODIS and CCI by considering that they are widely employed annual land cover datasets and other datasets

seldom provide annual land cover mapping results. Moreover, we also compared the CLCD with 30 m GlobeLand30 and FROM_GLC for a more comprehensive quality evaluation.

8) Line 243. The time span of GSW is different from that described in Line 324.

R: Thanks for your comment. We used an updated version (v1.2, available at code.earthengine.google.com) of GSW, which includes monthly water history from 1985 to 2018. Therefore, the time span is slightly longer in the line 324 than that in Pekel et al., (2016).

9) Line 244. "date" ->"data".

R: Done.

10) Line 256. Again, what does "dominant LC" signify here? Please specify it.

R: Thanks for your comment. The "dominant LC" herein refers to the land cover classes with a relatively large proportion in area. Accordingly, we revised the sentence in Line 256 as:

"*For the LCs with a relatively large proportion in area, such as cropland, forest and grassland*".

11) Line 297. Why were these three years chosen? How about other years?

R: Thanks for pointing out this issue. Accordingly, we also calculated the correlation coefficients of ISA fraction between CLCD-ISA and these three thematic datasets for each year (Fig. R1). It can be seen that the CLCD showed good consistency with the existing ISA products ($0.51 < R^2 < 0.83$). We therefore revised the manuscript as:

"*We first calculated ISA fractions within a 0.05°×0.05° spatial grid for each year and estimated the correlation coefficients between CLCD-ISA and the three thematic datasets to quantitatively demonstrate their agreement. Overall, the CLCD-ISA showed good consistency with the existing ISA products ($0.51 < R^2 < 0.83$), indicating the reliability of our CLCD products (Fig. 6)*"

[Figure]

**Figure R1. The correlation coefficients of ISA fraction between CLCD and three thematic datasets for each year.**

12) Line 324. "surfaec"->"surface".

R: Corrected.

13) Line 395. Users may be interested in the zoom-in maps of Fig. 13, but the spatial extent of some zoom-in maps seems inconsistent to the loss/gain images. It would be good to highlight the extent of zoom-in maps in the corresponding images.

R: Much obliged. According to your suggestion, we improved Fig. 13 by making the extent of zoom-in maps consistent with the corresponding images (Fig. R2).

[Figure]

**Figure R2. Typical land cover changes and conversions observed in China from 1985 to 2019:** (a) lake shrinkage in Wuhan City (30.543703° N, 114.296992° E); (b) cropland loss due to the urbanization in Shanghai City (31.214542° N, 121.490796° E); (c) deforestation induced by reclamation in northeast China (49.627007° N, 125.029041° E); (d) bare land converted to grassland in the Qira oasis, located at the southern fringe of the TAKLAMAKAN Desert (36.237807° N, 81.215592° E). The illustration is of Landsat images with false-colour combination (R: NIR, G: Red, B: Green) to enhance the grassland; (e) grassland loss due to the reclamation in northwest China (43.575843° N, 80.987817° E); (f) Siling Co Lake expansion in the Tibet Plateau (31.793826° N, 89.057216° E); (g) reclamation in the Aksu Prefecture (40.593062° N, 81.048109° E); (h) afforestation in Yunnan Province (26.518645° N, 103.613297° E); (i) Expansion of Chengdu City (30.666955° N, 104.068452° E); (j) grassland gain in the Mu Us Desert, Yulin City (38.509526° N, 109.660052° E).

14) Line 411. I notice that you have mentioned the uneven coverage of Landsat 5 for several times. Therefore, it is suggested to explicitly demonstrate or explain this issue.

R: Thanks for your suggestion. The year of first Landsat 5 acquisition in China varies significantly. For instance, the images were generally available over northern China around 1986, but were not available in the northwest until 1988 (Fig. R3). We therefore used all images before 1990 to generate the CLCD for 1985. Accordingly, we have revised the sentences as:

"*The year of first Landsat 5 acquisition in China varies significantly. For instance, the images were generally available over northern China around 1986, but were not available in the northwest until 1988 (Fig. S1). Therefore, we used all Landsat SR captured before 1990 to generate the CLCD of 1985 to minimize the influence induced by the availability of Landsat 5.*"

[Figure]

**Figure R3. Year of first Landsat 5 image over China.**

**References:**

Fritz, S., See, L., Perger, C., McCallum, I., Schill, C., Schepaschenko, D., Duerauer, M., Karner, M., Dresel, C., Laso-Bayas, J. C., Lesiv, M., Moorthy, I., Salk, C. F., Danylo, O., Sturn, T., Albrecht, F., You, L., Kraxner, F. and Obersteiner, M.: A global dataset of crowdsourced land cover and land use reference data, Sci. Data, 4, 170075, doi:10.1038/sdata.2017.75, 2017.

Pekel, J. F., Cottam, A., Gorelick, N. and Belward, A. S.: High-resolution mapping of global surface water and its long-term changes, Nature, 540(7633), 418–422, doi:10.1038/nature20584, 2016.

---

## Author Comment (AC2)

Referee #2

The manuscript named "30 m annual land cover and its dynamics in China from 1990 to 2019" proposed a multi-year land cover dataset derived from Landsat and SRTM DEM data by using a supervised classification method. Accuracy estimation showed the superiorities of the data. The manuscript was clearly written with a logical order that is easy to follow. I believe this manuscript can be considered to publish after some minor revision. Only some minor comments listed as below:

"GlobaLand30" should be "GlobeLand30"

R: Much obliged. Corrected.

"FROM_GLC" or "FROM-GLC"?

R: Many thanks. We replaced "FROM-GLC" with "FROM_GLC" throughout the manuscript.

References or an ablation experiment may be good for the proposed spatial-temporal filtering.

R: Thanks for your suggestion. An ablation experiment has been added in the revised manuscript. We calculated the confusion matrix for CLCD without spatial-temporal filtering. It was indicated that the overall accuracy of CLCD without the spatial-temporal filtering was dropped by 0.71% than that with spatial-temporal filtering (Table R1 & R2). This showed the effectiveness of our proposed spatial-temporal filtering. Following your suggestion, we also added the references and revised the manuscript:

"*We calculated the confusion matrix for CLCD without spatial-temporal filtering. It was indicated that the overall accuracy of CLCD without the spatial-temporal filtering was dropped by 0.71 than that with spatial-temporal filtering (Table S28 & S29). This showed the effectiveness of our proposed post-processing method.*"

**Table R1. Confusion matrix of CLCD without spatial-temporal filtering based on visually-interpreted test samples. PA and UA are abbreviations for the producer's accuracy and user's accuracy, respectively. Bold number represents the overall accuracy.**

| | Class | Cropland | Forest | Shrub | Grassland | Water | Snow/Ice | Barren | Impervious | SUM | UA (%) |
|---|---|---|---|---|---|---|---|---|---|---|---|
| | **CLCD without spatial-temporal filtering** | | | | | | | | | | |
| | Cropland | 955 | 62 | 4 | 91 | 5 | 0 | 8 | 14 | 1139 | 83.85 |
| | Forest | 72 | 962 | 9 | 56 | 0 | 0 | 0 | 1 | 1100 | 87.45 |
| | Shrub | 21 | 65 | 36 | 32 | 0 | 0 | 2 | 0 | 156 | 23.08 |
| **Visually-** | Grassland | 79 | 57 | 16 | 983 | 1 | 9 | 71 | 2 | 1218 | 80.71 |
| **interpreted** | Water | 10 | 3 | 0 | 2 | 109 | 1 | 2 | 2 | 129 | 84.50 |
| **samples** | Snow/Ice | 0 | 0 | 0 | 2 | 0 | 79 | 5 | 0 | 92 | 85.87 |
| | Barren | 5 | 0 | 0 | 351 | 6 | 11 | 1028 | 5 | 1413 | 72.75 |
| | Impervious | 62 | 2 | 0 | 9 | 0 | 0 | 1 | 142 | 216 | 65.74 |
| | SUM | 1197 | 1173 | 62 | 1510 | 119 | 101 | 1114 | 176 | 5463 | |
| | PA (%) | 79.78 | 82.01 | 58.06 | 65.10 | 91.60 | 78.22 | 92.28 | 80.68 | | **78.60** |

**Table R2. Confusion matrix of CLCD with spatial-temporal filtering based on visually-interpreted test samples. PA and UA are abbreviations for the producer's accuracy and user's accuracy, respectively. Bold number represents the overall accuracy.**

| | Class | Cropland | Forest | Shrub | Grassland | Water | Snow/Ice | Barren | Impervious | SUM | UA (%) |
|---|---|---|---|---|---|---|---|---|---|---|---|
| | **CLCD with spatial-temporal filtering** | | | | | | | | | | |
| | Cropland | 963 | 61 | 2 | 83 | 5 | 0 | 5 | 20 | 1139 | 84.55 |
| | Forest | 68 | 976 | 5 | 48 | 0 | 1 | 0 | 2 | 1100 | 88.73 |
| | Shrub | 19 | 68 | 38 | 29 | 0 | 0 | 2 | 0 | 156 | 24.36 |
| **Visually-** | Grassland | 72 | 72 | 15 | 982 | 1 | 9 | 63 | 4 | 1218 | 80.62 |
| **interpreted** | Water | 9 | 3 | 0 | 3 | 108 | 1 | 3 | 2 | 129 | 83.72 |
| **samples** | Snow/Ice | 0 | 0 | 0 | 3 | 0 | 84 | 5 | 0 | 92 | 91.30 |
| | Barren | 5 | 0 | 0 | 347 | 5 | 13 | 1036 | 7 | 1413 | 73.32 |
| | Impervious | 58 | 3 | 0 | 8 | 0 | 0 | 1 | 146 | 216 | 67.59 |
| | SUM | 1194 | 1183 | 60 | 1503 | 119 | 108 | 1115 | 181 | 5463 | |
| | PA (%) | 80.65 | 82.50 | 63.33 | 65.34 | 90.76 | 77.78 | 92.91 | 80.66 | | **79.31** |